# Comparative Ergonomic Study Examining the Work-Related Musculoskeletal Disorder Symptoms of Taiwanese and Thai Workers in a Tape Manufacturing Factory

**DOI:** 10.3390/ijerph20042958

**Published:** 2023-02-08

**Authors:** Yi-Lang Chen, Wen-Hua Luo

**Affiliations:** 1Department of Industrial Engineering and Management, Ming Chi University of Technology, New Taipei 24301, Taiwan; 2Seal King Industrial Corporation, Taoyuan 33044, Taiwan

**Keywords:** tape manufacturing, logistic regression, musculoskeletal disorder symptoms, Nordic Musculoskeletal Questionnaire (NMQ), risk factors

## Abstract

This study surveyed 114 Taiwanese and 57 Thai workers in a tape manufacturing factory in Taiwan and evaluated their symptoms of work-related musculoskeletal disorder (WMSD) and associated risk factors by using the revised Nordic Musculoskeletal Questionnaire. Task-appropriate biomechanical and body load assessment tools were also employed to examine biomechanical and body load during four specified daily tasks. The results indicated that the prevalence of discomfort symptoms in any body part within one year was 81.6% for the Taiwanese workers and 72.3% for the Thai workers. The body part in which the Taiwanese workers most frequently experienced discomfort was the shoulders (57.0%), followed by the lower back (47.4%), the neck (43.9%), and the knees (36.8%); where the Thai workers most frequently experienced discomfort was the hands or wrists (42.1%), followed by the shoulders (36.8%) and the buttocks or thighs (31.6%). These locations of discomfort were associated with task characteristics. Heavy-material handling (>20 kg) more than 20 times per day was the most significant risk factor for WMSDs for both groups, and this task must thus be urgently improved. We also suggest that providing wrist braces for Thai workers may assist in alleviating their hand and wrist discomfort. The biomechanical assessment results indicated that the compression forces acting on the workers’ lower backs exceeded the Action Limit standard; administrative controls must thus be instituted for two heavy-material handling tasks. In the factory, some tasks and workers’ movements when completing these tasks must be assessed and improved immediately by using appropriate tools. Although the Thai workers were engaged in more physically demanding tasks, their WMSDs were milder than those of the Taiwanese workers. The results of the study can serve as references for the prevention and reduction of WMSDs in local and foreign workers in similar industries.

## 1. Introduction

Musculoskeletal disorders (MSDs) are injuries or disorders of the muscles, nerves, tendons, joints, peripheral nerves, vascular system, and spinal discs that can cause sprains, pain, and inflammatory and degenerative changes [1,2]. Work-related musculoskeletal disorders (WMSDs) are MSDs that are caused or exacerbated by work, work-related factors, or individual reasons [3,4]. WMSDs account for a significant proportion of disorders and disability cases, indicating that WMSDs not only cause immense physical and mental health strain but also impose a heavy burden on health care services [5,6,7].

Workers in different industries experience differing levels of WMSD severity and associated factors. Investigations have focused on specific industries and tasks [8,9,10,11,12,13,14]. The manufacturing industry has drawn considerable attention because of the various factors in this industry that can induce WMSD, including heavy-load lifting, awkward work postures, high exposure, intense vibration, and extreme environmental situations [15]. In Taiwan, more than 98% of companies in the manufacturing industry are small or medium-sized enterprises. Because of cost, manual tasks remain the main means of production. Thus, the WMSDs of on-site employees in the manufacturing industry warrant evaluation. Overall, the WMSDs of manufacturing employees in Taiwan are more severe than those of workers in other industries [9].

On the basis of the cost economy, which is a strategy adopted in many countries, the manufacturing industry in Taiwan has recruited large numbers of foreign workers from Southeast Asia since 1989 to work in on-site production and fill a labor gap. To date, approximately 500,000 foreign workers have been employed in Taiwan in various manufacturing industries [16]. However, this phenomenon has led to negative occupational health outcomes for international migrant workers. Such outcomes have been highlighted and investigated in various countries including South Korea [17], Singapore [18], Denmark, the United Kingdom and the Netherlands [19], Australia [20], Spain [21], Canada [22], and Taiwan [23]. Orrenius and Zavodny [24] reported that immigrants were more likely to work in jobs where their safety is at higher risk than local workers, partly because of immigrant workers’ characteristics, such as their lower levels of language ability and educational attainment. Moyce and Schenker [25] observed that worldwide, immigrant workers are more likely to have adverse occupational exposures and working conditions, which leads to poor health outcomes, workplace injuries, and occupational fatalities. Recently, Hargreaves et al. [26] systematically surveyed 36 related studies and concluded that international migrant workers are at considerable risk of work-related ill health and injury and that their health needs are critically overlooked in both research and employment policy. Governments, policymakers, and employers must enforce and improve occupational health and safety measures and implement appropriate strategies to meet the care needs of this key working population.

Among all industries in which foreign workers are engaged, the construction industry has attracted the most research attention [17,27,28,29]. However, according to the results of a survey conducted by Taiwan’s Occupational Safety and Health Administration [30], the manufacturing industry accounts for the highest percentage of WMSDs (29.2%), whereas the construction industry accounts for the second highest percentage (19.4%). This implies that the WMSDs of manufacturing industry employees are of particular interest. In addition, although many aspects of the WMSDs of foreign workers have been examined, our review of the literature revealed few comparative studies. This motivated us to conduct the cross-sectional study. This study surveyed 114 Taiwanese employees and 57 Thai employees of a tape manufacturing factory in northern Taiwan and used the revised Nordic Musculoskeletal Questionnaire (NMQ) to evaluate the prevalence of discomfort in various body parts and potential risk factors. Biomechanical (3D Static Strength Prediction Program [3DSSPP]) and body workload assessment tools (Rapid Upper Limb Assessment [RULA] and Ovako Working Posture Analysis System [OWAS]) were employed in accordance with the characteristics of relevant tasks to examine the degree of hazards in four typical and frequently performed tasks.

## 2. Methods

### 2.1. Participants

This study investigated a medium-sized tape manufacturing factory located in Taoyuan, Taiwan. The factory employs approximately 200 full-time workers, including 65 Thai workers. Excluding sales and marketing personnel and employees who worked at the factory for less than one year, a total of 186 NMQ questionnaires were distributed, and consequently, 114 and 57 valid questionnaire responses from Taiwanese and Thai workers were collected, respectively, with an effective rate of 91.9%. The NMQ was administered on a one-on-one basis through two language versions, namely Chinese and Thai versions. Written informed consent was obtained from all participants, and the study protocol received approval from the Ethics Committee of Chang Gung Memorial Hospital, Taiwan.

### 2.2. Task Description

The tape manufacturing process typically comprises the following steps: (1) receipt and inspection of materials, (2) storage, (3) gluing, (4) placement, (5) processing, (6) packaging, and (7) storage and delivery. Many potential risk factors for musculoskeletal injuries are associated with these tasks, such as the handling of heavy raw materials, repetitive ladling operations, awkward pushing and pulling postures, and rapid packaging actions.

During our field observation, four daily heavy-duty tasks were reported by the workers and were evaluated in terms of body load by using the appropriate tools. The following four cases, one for each task, were investigated: Case A was the lowering of inventory, in which the workers lowered material weighing approximately 65 kg onto a pallet. Case B was the pulling of raw material, in which the workers pulled raw material weighing more than 200 kg (up to 1000 kg) backward by using a hydraulic trolley. Case C was the ladling of glue, in which the workers used a pail to transfer glue to a tank; they usually scoop 1000 kg of glue every day and thus perform the action up to 500 times a day. Case D was the feeding of material, in which the workers bent at the trunk and performed a stomping motion when changing material under the oven. These cases are depicted in Figure 1A–D, respectively. The postures presented in Figure 1A,B,D were the most difficult positions reported by the workers and were also the heaviest tasks or poorest postures identified by the two professional ergonomists. However, Figure 1C shows a series of repetitive actions for ladling glue.

### 2.3. NMQ

The NMQ was designed on the basis of in-depth interviews with Taiwanese and Thai workers, field observations, and a review of the literature. The questionnaire was divided into three sections. The first section collected basic personal information, and the second section focused on job characteristics, including individual factors, daily primary tasks and task requirements, previous injuries, and usage of personal assistive devices. The task-related questionnaire items were adapted from the questionnaire developed by the Institute of Labor, Occupational Safety, and Health (ILOSH), Taiwan [9].

In the third section of the questionnaire, NMQ items were used to investigate symptoms of musculoskeletal discomfort in the workers. The NMQ is a general questionnaire that classifies musculoskeletal discomfort and symptoms of disorders on the basis of nine major body parts. This tool enables investigators to discriminate between sites of discomfort and injury and to closely examine symptoms specific to certain sites. A map facilitating the identification of particular body parts was used [31]. Takekawa et al. [32] concluded that the NMQ is the most effective instrument for identifying whether respondents experience chronic or recurring lower back pain. Deakin et al. [33] determined that this self-report questionnaire has reliability and validity ranging from 77% to 100% and from 80% to 100%, respectively. The questionnaire was also determined to be suitable for use in Taiwanese populations [10,12,34,35].

The content validity and reliability of the revised NMQ were assessed by three experts: a senior occupational safety and health executive at a manufacturing company and two occupational health professionals [36]. Reliability was examined using the test–retest method (involving data from 25 and 14 Taiwanese and Thai workers, respectively) and by calculating the Pearson product–moment correlation coefficient. The interval between the first and second tests was 2 weeks. The correlation coefficients (r) ranged from 0.838 to 0.941 for all questionnaire items, which were designed to elicit various answers at specific frequencies.

### 2.4. 3DSSPP

When the workers performed the heavy-material handling tasks depicted in Figure 1A,B, the L4/L5 disc compression forces were estimated using the 3DSSPP (version 7.1.3), which was developed by the University of Michigan’s Center for Ergonomics. Joint angles, anthropometric data, and strength values were input into the program to estimate the lumbar compression forces. In the present study, anthropometric data collected from the Taiwanese and Thai workers were input into the system to account for anthropometric factors in the population. To aid posture entry, an inverse kinematics algorithm was developed using 3DSSPP data; this algorithm was based on the handling postures of people manipulating loads by using known hand positions. However, the 3DSSPP is most useful in the analysis of slow movements executed during heavy-material handling tasks because the biomechanical computations assume that the effects of acceleration and momentum are negligible [37]. In this study, the material loads were considered to be forces exerted through the workers’ hand manipulation, and the L4/L5 disc compression forces were thus calculated.

### 2.5. RULA

RULA was developed to provide a rapid assessment of the loads on the musculoskeletal system of operators due to posture, muscle function, and the forces they exert [38]. Using the RULA method, there should be an effort to reduce subjectivity by measuring the position angles of segments of the musculoskeletal system. Having multiple photographs of particular postures, an assessment can be made for each position, and by measuring the time taken to hold each position, the level of exposure to a given level of risk during work can be estimated. Of the various WMSD assessments that are frequently employed, the RULA has been widely applied in many industries, and RULA results have been verified to be significantly associated with postural load criteria such as discomfort symptoms [39]. In this method, group A comprises the upper arms, lower arms, and wrists, and group B comprises the neck, trunk, and legs. The score for group A postures and group B postures and the scores for static muscle work and force are summed as appropriate to obtain a C score (upper limbs) and a D score (neck, trunk, and legs). The C and D scores are then combined and tabulated to obtain a Grand Score. The Grand Score is used to assign the observed posture to an Action Level that indicates the required intervention. The RULA has been demonstrated to be reliable in adults [38].

In this study, the RULA was used to evaluate the body load of the workers when they performed the repetitive ladling task depicted in Figure 1C. During the assessment, two professional ergonomists who were familiar with RULA first observed and determined the work cycle, photographed the ladling postures from different views, and then proceeded with the subsequent assessment based on the RULA procedure, as suggested by McAtamney and Corlett [38]. Finally, the RULA score was calculated, and the level of MSD risk was then determined. The result was also checked with the online RULA software (https://www.rula.co.uk (accessed on 26 January 2023) Osmond Ergonomics, Osmond Group Limited, Wimborne, UK).

### 2.6. OWAS

The OWAS was developed in Finland by Ovako Metals Oy AB, a leading European producer of steel bars and profiles [40]. The OWAS method was intended to identify the frequency of and time spent in the postures adopted when performing a certain task, to study and evaluate related situations, and to then recommend corrective actions. The OWAS identifies the most typical back (four postures), arm (three postures), and leg postures (seven postures) adopted and the weight of the load handled (three categories), with there being 252 (4 × 3 × 7 × 3) possible combinations. Each posture assumed by a worker is assigned a four-digit code in accordance with the classification of the identified postures for each part of the body and the load. However, the OWAS method has been continuously modified and extended to match the varying real task conditions, and is applied in many industries [34,41,42,43].

In this study, the awkward working posture adopted by the workers when feeding material, illustrated in Figure 1D, was examined using the OWAS method. Similar to the assessment in Case C, two professional ergonomists assessed Case D. The observed worker had an average stature among the workers in the factory. The ergonomists recorded the whole cycle of the feeding material task and determined the targeted posture, as shown in Figure 1D. Because the working posture was nearly static, the back, arm, and leg postures were relatively easy to be identified. Once the whole body posture of the worker and the load handled were coded based on the coding system of OWAS, the risk level was then ascertained.

### 2.7. Statistical Analysis

Statistical analyses were conducted using SPSS software (version 22.0; IBM, Armonk, NY, USA), with *p <* 0.05 indicating statistical significance. Questionnaire data were examined using descriptive statistics and logistic regression. Logistic regression analysis was applied to determine the potential risk factors associated with the discomfort symptoms in each body site of the participants. The factors surveyed by the questionnaire covered personal information, job characteristics (individual factors, daily primary tasks, task requirements), previous injuries, and the usage of personal assistive devices. The odds ratio (OR) was employed to compare the relative odds of the occurrence of certain variables. The Pearson product–moment correlation coefficient r was used to explore the test–retest reliability of the NMQ responses for the preselected Taiwanese and Thai workers. In addition, the chi-square test was carried out to determine the significance of the difference in the prevalence of the discomfort of various body sites between the two worker groups.

## 3. Results

### 3.1. Demographics and Task Characteristics

Among the 186 respondents, 15 were excluded who did not meet the recruitment criteria, and valid data from 114 Taiwanese and 57 Thai workers were finally analyzed. Table 1 presents the basic data for the workers. The mean (standard deviation) age, stature, and body mass for the Taiwanese workers were 38.5 (8.7) years, 166.0 (7.8) cm, and 66.5 (15.2) kg, respectively, and those for the Thai workers were 36.5 (7.5) years, 166.5 (5.9) cm, and 67.1 (10.1) kg, respectively. Except for job tenure, there was no difference in the basic data between the two worker groups. Table 2 details the demographics and task characteristics of the 171 workers; the Taiwanese and Thai respondents included 33 and 4 women, respectively. Among the respondents, surprisingly, two-fifths of the Taiwanese workers had no exercise habits, and less than 10% of the Thai workers did. The proportions of workers who regularly smoked (Taiwanese, 78.0%; Thai, 49.1%) and consumed alcohol (67.5%; 85.9%) were relatively high. Among the respondents, 74 Taiwanese and 25 Thai workers had an MSD, and 45.0% and 28.0% reported that they had not fully recovered from the MSD.

The task characteristics of workers in the tape manufacturing factory are detailed in Table 2. For both the Taiwanese (42.1%) and Thai (45.6%) workers, roll production was the primary task, followed by computer operations for the Taiwanese workers (35.1%) and gluing for the Thai workers (31.6%). When not performing sitting-based tasks, the Thai workers stood, walked, assumed awkward postures, and performed heavy-duty handling more frequently and for longer periods of time than the Taiwanese workers did. Most of the Thai workers used personal protective equipment, such as back support, when carrying materials for comfort purposes. Compared with the Taiwanese workers (82.6%), however, fewer Thai workers used wrist supports (12.3%).

### 3.2. WMSDs and Risk Factors

Table 3 summarizes the NMQ results. The prevalence of musculoskeletal discomfort symptoms within one year in the Taiwanese and Thai workers was 81.6% and 77.3%, respectively. The Taiwanese workers reported discomfort primarily in their shoulders (57.0%), lower back (47.4%), neck (43.9%), and knees (36.8%), whereas the Thai workers reported discomfort in mostly their hands or wrists (42.1%), shoulders (36.8%), and buttocks or legs (31.6%). The prevalence of discomfort in the shoulders, neck, lower back, and knees of the Taiwanese workers was 20% higher than that for the Thai workers. Most of the differences in the prevalence of discomfort between the two groups for various body sites were significant using the chi-square test (Table 3). Further analyses indicated that the main discomfort symptom in these body parts was an aching pain (average 76.2%) that slightly reduced the workers’ ability to work (approximately 50%), and approximately one-third of the respondents reported that such pain occurred almost weekly. Most respondents ignored the discomfort or left them untreated (63.8%). A high proportion of the respondents considered the discomfort to have been induced completely by work (average 83.8%).

Table 4 presents the results of the logistic regression analysis conducted to identify risk factors for discomfort in the three body parts for which the prevalence of discomfort was highest in the two worker groups. The results indicated that personal habits, sustained working time, and awkward working postures as well as frequent material handling were the main causes of WMSD symptoms.

### 3.3. Body Load Assessments of Specified Cases

In this study, four specified cases (Figure 1) were assessed using the matched-analysis tools. Cases A and B were analyzed using the 3DSSPP, whereas Cases C and D were analyzed using the RULA and OWAS, respectively. The results revealed that the compression forces acting on the L4/L5 discs were 4607 and 5843 N when performing the tasks in Cases A and B, respectively (Figure 2). The load on the lower back exceeded the Action Limit (AL; 3400 N) for the weight of a given lifting task, as recommended by the National Institute for Occupational Safety and Health (NIOSH), and administrative controls must therefore be intervened [44].

For Case C, the calculated Grand Score was 7; the corresponding Action Level of 7 indicates that the worker is working in the most unfavorable posture and has an imminent risk of injury. Changes must be made in the near future to prevent such an injury [38]. For Case D, the OWAS assigned the worker a code of 4341; the corresponding Action Category was 4, which indicates that the posture must be corrected immediately [40].

## 4. Discussion

In our analyses, the prevalence of musculoskeletal discomfort in any body part within a year was 81.6% and 72.3% for Taiwanese and Thai workers, respectively. Although the work tasks of the two groups were different and the Thai workers spent more time performing physical actions (e.g., standing and walking) [24], we observed many similarities. This implies that the two worker groups may have different levels of health status, which affected their self-reported discomfort symptoms suffered from the tasks and must be taken into account during task redesign and rearrangement.

As presented in Table 3, the cohort data were also compared with those obtained from other populations in Taiwan [9]. As shown in the table, the Taiwanese workers were more likely to experience discomfort in their neck, shoulders, lower back, and knees, whereas the Thai workers were more likely to experience discomfort in their hands or wrists and buttocks or legs in comparison with the bodily discomfort experienced by workers in similar industries [9]. However, the overall prevalence of musculoskeletal discomfort within one year for Taiwanese (81.6%) and Thai workers (72.3%) was higher than that recorded in that systematic study (49.8% for manufacturing industries). This implies that the manufacturing pattern may be the main cause of WMSDs due to differences in job characteristics, resulting in different degrees of the prevalence observed in musculoskeletal discomfort symptoms.

For Taiwanese workers, the prevalence of discomfort symptoms within one year in the shoulders, neck, lower back, and knees in the investigated sample was 14.0%, 18.7%, 19.0%, and 24.2%, respectively, higher than the prevalence in similar industries (Table 3). These results indicate that in the tape manufacturing industry, the potential risks of WMSDs are higher than those in other manufacturing industries. The logistic regression results (Table 4) indicated that the handling of heavy objects (>20 kg) more than 20 times per day had a definite impact on discomfort in various parts of the body (all *p* < 0.05), such as in Cases A and B (Figure 1). The object weight (i.e., >20 kg) is close to the load constant (23 kg), which refers to the maximum recommended weight for lifting at the standard lifting location under optimal conditions, as suggested by NIOSH [45], and in line with the work situation of Asians [46,47]. This typical risk factor has been recognized in the literature, and Keyserling [15] argued that the heavy load accompanied by awkward working postures and high levels of repetition are the main causes of WMSDs. In our study, sex, back movement (i.e., deep bending and twisting), sitting time, and age also contributed to the discomfort experienced in the shoulders, lower back, neck, and knees, respectively, of Taiwanese workers. 

Multiple studies have reported that female workers generally experience more shoulder discomfort than male workers when performing various tasks [48,49,50]; our results are consistent with those reported results. However, in the present study, standing and walking for a long time (>4 h per day) also led to shoulder discomfort, possibly because these actions were accompanied by various upper-limb handling tasks in the tape manufacturing factory. Regarding the effect of exercise habits on shoulder discomfort, similar results have been reported in other studies [12,51]. In our study, 35.1% of Taiwanese workers engaged in computer operations on a daily basis. This task required long hours of sitting (>4 h per day), resulting in neck discomfort [52,53]. Cagnie et al. [54] indicated that women had an almost two-fold higher risk of experiencing neck pain when working in an office as compared with men. Cohen and Hooten [55] also noted that women experienced neck pain more frequently than men. Because several male workers who performed computer operations were included in our study, sitting time, not sex, may therefore be a significant risk factor for neck discomfort. A significant relationship between working time spent sitting and neck pain was reported by Cagnie et al. [54]. Our analyses also indicated that tobacco smoking was a factor affecting neck discomfort. The etiology of work-related neck disorders is regarded as multidimensional, with such disorders associated with and influenced by a complex array of individual, physical, and psychosocial factors [12,54], such as tobacco smoking. In addition, demographic factors such as age are associated with discomfort in the knee [56,57], which we observed in the Taiwanese workers in our study.

Although many studies have investigated the WMSDs of workers in different countries, few studies have compared the WMSDs of workers of different nationalities in the same workplace. This study compared the WMSDs of Taiwanese and Thai workers in a single tape manufacturing factory. The prevalence of discomfort in the shoulders, neck, lower back, and knees was higher in the Taiwanese workers than the Thai workers, whereas the prevalence of discomfort in the hands or wrists and buttocks or legs was higher in the Thai workers (Table 3; +13.2% and +14.1%, respectively). Wu et al. [58] proposed that the risk of occupational injuries for foreign workers was lower than that for local Taiwanese workers. However, the risks of occupational injuries and WMSDs in these two groups are inherently different. A possible reason is that, in general, only healthy foreign workers come to Taiwan; this may result in a preselection effect and merits further investigation. The data presented in Table 2 indicate that, compared with the Taiwanese workers, the Thai workers mostly had exercise habits (91.2%), smoked less, and had fewer historical injuries but consumed more alcohol. None of the Thai workers performed computer operations, but many performed gluing. However, the proportion engaged in roll production (45.6%) was almost the same as that of the Taiwanese workers (42.1%). As reported in related surveys [24,26], we observed that the foreign workers mostly performed unskilled physical tasks in this study. This may be the reason that most of the Thai workers used back supports (93.5%); however, they seldom used wrist supports (12.3%). The results of the logistic regression can be explained on the basis of this information. The handling of heavy objects (>20 kg) more than 20 times per day was the main cause of hand or wrist and shoulder discomfort, whereas long periods of standing and carrying over distances induced buttock or leg discomfort. Back support use and wrist support nonuse may have directly reduced the prevalence of lower back discomfort and increased hand or wrist discomfort, respectively. As detailed in Table 4, these factors were not selected as risk factors and may apply to limited samples. 

In addition to performing the NMQ investigation, this study also analyzed four typical daily tasks in tape manufacturing (Figure 1) by using multiple assessment tools (i.e., the 3DSSPP, RULA, and OWAS). The results indicated that the body loads exerted during these tasks were likely to cause WMSDs and thus must be reduced immediately. Notably, the highest prevalence of discomfort symptoms for Thai workers was observed in the upper extremities (hands or wrists, 42.1%; shoulders, 36.8%). This may also be partly related to those Thai workers performing more ladling tasks (Case C) than the Taiwanese workers, and that they also seldom used wrist supports. The manual ladling task is extremely inefficient and the RULA result also showed an imminent risk of injury. The suggested improvement is to change the manual ladling by equipping a liftable and tiltable trough for the operation. We also suggest that providing wrist supports for the Thai workers and encouraging their use may assist in alleviating hand and wrist discomfort. For both worker groups, the handling of heavy objects (>20 kg) more than 20 times per day was the most dominant risk factor for WMSDs. Reducing the frequency of this task through job redesign (ex., changing from lifting to pushing and pulling for Case A and changing the way of material feeding for Case D) or the supplying of appropriate auxiliary delivery devices (e.g., electric pallet truck for Case B) could significantly reduce the prevalence of WMSD symptoms in various body parts.

In our study, the 3DSSPP was used to estimate the forces on spinal discs exerted in Cases A and B. Although the estimated forces did not reach the maximum permissible limit (MPL; 6400 N) stated in the NIOSH lifting guides [44], biomechanical computations assume that the effects of acceleration and momentum are negligible [37]. This indicates that the 3DSSPP is a static-based model. From our on-site observations, we determined that the workers must generate sufficient initial acceleration to complete these tasks successfully (Case A, lift off; Case B, overcome static friction). Studies have verified that static biomechanical models underestimate the actual lumbar spine force [59], sometimes by a factor of more than one [60]. A recent study compared the static and dynamic biomechanical models and also found that failing to account for body motions can underestimate net joint moments by approximately 90% of the static estimates during lifting [61], and this would considerably underestimate the spinal compression forces. When performing Case A and B tasks, the actual forces on the workers’ lower back may have exceeded the MPL [44], which is extremely harmful to the lower back. Because the tasks performed in Cases C and D reached the hazard levels in the RULA and OWAS, respectively, these tasks must also be redesigned. Obviously, the upper body loading was concerned about Case C, and 3DSSPP was not applicable. In Case D, the workers adopted awkward working postures because of the restricted size of the workspace. However, when the task was evaluated using the 3DSSPP, the L4/L5 compression force was calculated to be only 3260 N, which does not exceed the AL because of the low load on the hands. By contrast, the OWAS classified this task as a hazardous operation. Therefore, tasks with different characteristics require the adoption of appropriate tools for accurate risk assessment [39,62].

Several limitations of this study must be highlighted. The questionnaire survey was limited to only 114 Taiwanese and 57 Thai workers in Taiwan. In this study, the investigated foreign workers were all Thai because the case factory only recruited Thai people as foreign workers. In addition to Thailand, the main countries from which foreign workers are recruited to work in Taiwan are Indonesia, Vietnam, and the Philippines [16]. The current findings thus cannot be extensively generalized to foreign workers of different nationalities. In this study, the discomfort symptoms were all subjectively self-reported by the workers, and no medical physical examination supported the results. When assessing the lower back loadings of workers in Cases A and B, the estimated values were not the true compression forces on the L4/L5 disc but the values assigned to the compression forces by applying the 3DSSPP model. The differences between the estimated and the true values were due to components of measurement uncertainty, which may include the quality of the digitization/modeling of postures and loads in the digital human model. Furthermore, the job assignments of the two groups of workers could not be clearly distinguished, and caution must be exercised when exploring cause-and-effect in WMSDs.

## 5. Conclusions

This study compared the WMSDs and risk factors of 114 Taiwanese workers and 57 Thai workers in a tape manufacturing factory in Taiwan. Four typical daily operations were assessed using the 3DSSPP, RULA, or OWAS, depending on the task’s characteristics. The results revealed that the prevalence of discomfort in any body part within one year was 81.6% for the Taiwanese workers and 72.3% for the Thai workers. Although the Thai workers were engaged in more physically demanding tasks than the Taiwanese workers were, their WMSDs were milder. The body parts in which the Taiwanese workers experienced the most discomfort were the shoulders, lower back, neck, and knees, and those in which the Thai workers experienced the most discomfort were the hands or wrists, shoulders, and buttocks or thighs. These body parts affected by discomfort were associated with the characteristics of the workers’ tasks. For both worker groups, handling heavy objects (>20 kg) more than 20 times a day was the primary risk factor for WMSDs, and this task must therefore be assessed and redesigned. In addition, the assessment results of four typical daily operations revealed the potential risks of WMSDs. These results can be referenced in the redesign and rearrangement of these tasks.

## Figures and Tables

**Figure 1 ijerph-20-02958-f001:**
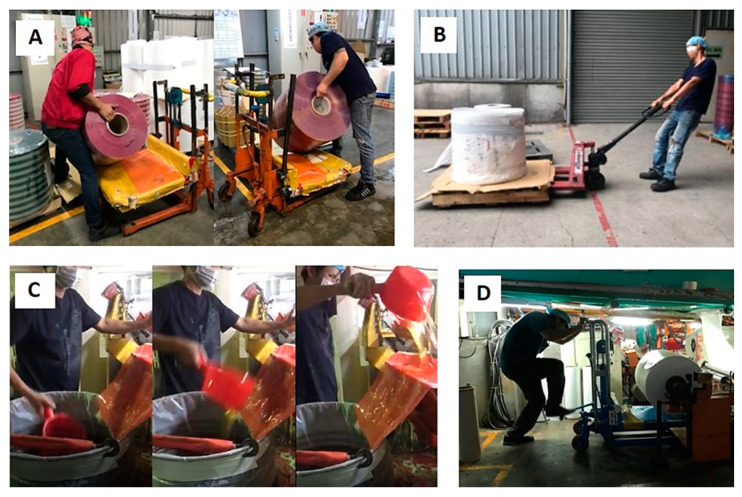
Four daily heavy-duty tasks performed during the tape manufacturing process: (**A**) lowering of inventory; (**B**) pulling of raw material; (**C**) ladling of glue; and (**D**) feeding of material.

**Figure 2 ijerph-20-02958-f002:**
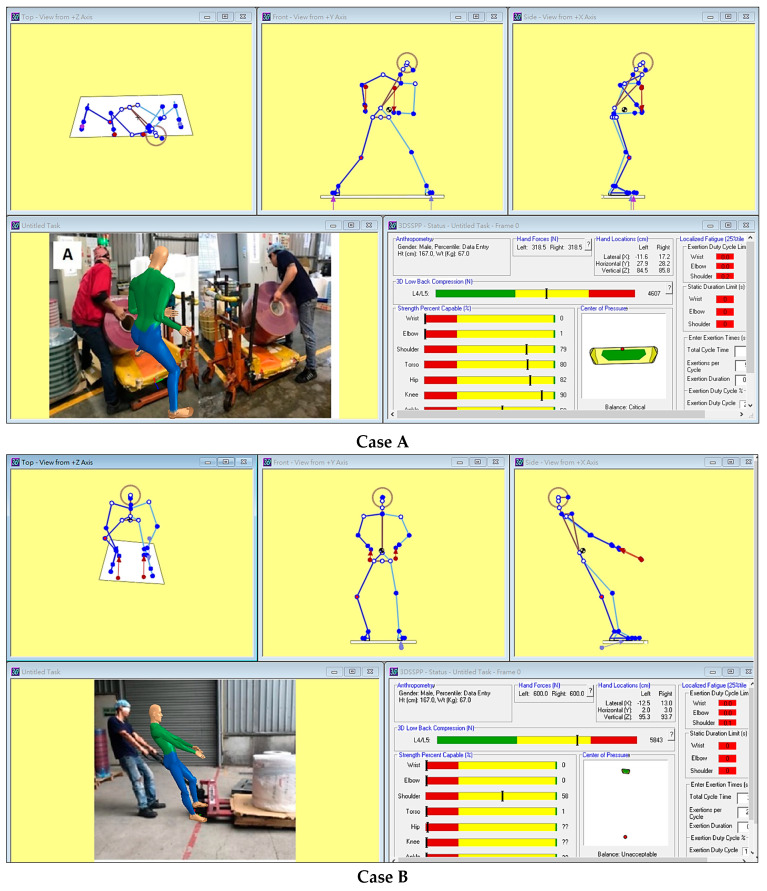
Loads on the lower back in Cases A and B, as determined using the 3D Static Strength Prediction Program.

**Table 1 ijerph-20-02958-t001:** Basic information of the 114 Taiwanese and 57 Thai workers.

	Taiwanese (*n* = 114)		Thai (*n* = 57)	
Item	Mean	Standard Deviation	Mean	Standard Deviation
Age (years)	38.5	8.7	36.5	7.5
Stature (cm)	166.0	7.8	166.5	5.9
Body mass (kg)	66.5	15.2	67.1	10.1
Job tenure (years)	6.3	7.3	3.6	3.8
Workdays per week (days)	5.1	0.3	5.2	0.4
Sleeptime per day (hours)	7.0	0.7	7.2	1.0

**Table 2 ijerph-20-02958-t002:** Basic information and task characteristics of the 114 Taiwanese and 57 Thai workers.

Variable	Category	Taiwanese, *n* (%)	Thai, *n* (%)
Individual factors			
Gender	Male Female	81 (71.1) 33 (28.9)	53 (93.0) 4 (7.0)
Dominant hands	Right Left	82 (71.9) 21 (18.4)	74 (66.7) 12 (21.0)
	Others	11 (9.7)	14 (12.3)
Exercise habits	None Sometimes or regular	46 (40.3) 68 (59.7)	5 (8.8) 52 (91.2)
Tobacco smoking	Yes	89 (78.0)	28 (49.1)
Alcohol drinking	Yes	77 (67.5)	49 (85.9)
Historical injuries			
Musculoskeletal disorders	Yes	74 (64.9)	25 (43.9)
(from *n* = 74 or 25)	Neck/shoulder pain	26 (35.1)	2 (8.0)
	Lower back pain	18 (24.3)	7 (28.0)
	Knee pain Scoliosis	9 (12.2) 7 (9.6)	4 (16.0) 0 (0)
Full recovery (from *n* = 74 or 25)	No	33 (45.0)	7 (28.0)
Daily primary tasks	Computer operation	40 (35.1)	0 (0)
	Roll production	48 (42.1)	26 (45.6)
	Gluing	11 (9.6)	18 (31.6)
	Cutting	9 (7.9)	9 (15.8)
	Packaging	6 (5.3)	4 (7.0)
Task requirements (per day)			
Sitting time	>4 h	42 (36.8)	4 (7.0)
Standing time	>4 h	41 (36.0)	51 (89.5)
Walking time	>4 h	30 (26.3)	49 (86.0)
Deep bending	>20 times	46 (40.4)	35 (61.4)
Twisting the waist	>20 times	49 (42.9)	30 (52.6)
Handling materials (>20 kg)	>20 times	43 (37.7)	36 (63.2)
Carrying distance	>5 m	78 (68.4)	28 (49.1)
Assistive devices usage			
Assistive for carrying (from *n* = 77 or 50)	Yes Cart Hydraulic trolley Pallet truck	77 (67.6) 23 (30.0) 13 (16.9) 16 (20.8)	50 (87.7) 7 (14.0) 29 (58.0) 1 (2.0)
Personal protective equipment (from *n* = 23 or 46)	Yes Back support Wrist protector	23 (20.2) 17 (73.9) 19 (82.6)	46 (80.7) 43 (93.5) 6 (12.3)

**Table 3 ijerph-20-02958-t003:** Prevalence of musculoskeletal discomfort in various body parts of Taiwanese and Thai manufacturing industry workers and of workers in other industries (unit: percentage).

	The Present Study			ILOSH [9]	
Body Parts	Taiwanese (*n* = 114)	Thai (*n* = 57)	Difference between Two Groups	Entire Working Population (*n* = 17,757)	Manufacturing Industry (*n* = 3401)
Neck	43.9	17.5	−26.4 ***	32.3	29.7
Shoulders	57.0	36.8	−20.2 ***	41.3	38.3
Upper back	25.4	28.1	2.7	22.3	20.8
Elbows	19.3	12.3	−7.0 *	20.5	19.6
Lower back	47.4	22.8	−24.6 ***	31.0	28.4
Hands/wrists	28.9	42.1	13.2 **	26.5	25.5
Buttocks/legs	17.5	31.6	14.1 **	11.8	10.3
Knees	36.8	15.8	−21.0 ***	16.9	12.6
Ankles	18.4	28.1	9.7 **	14.6	12.2

Notes: * *p* < 0.05, ** *p* < 0.01, *** *p* < 0.001, the difference was examined using the chi-square test.

**Table 4 ijerph-20-02958-t004:** Risk factors significantly associated with musculoskeletal disorder symptoms (prevalence > 30%).

	Body Parts (Prevalence %)	Risk Factors	Category	*n*	OR	95% CI
Taiwanese workers (*n* = 114)	Shoulders (57.0%)	Se Exercise habits Handling materials (>20 kg) Standing time Walking time	Male Female None Sometimes or regular ≤20 times per day >20 times per day ≤4 h per day >4 h per day ≤4 h per day >4 h per day	81 33 46 68 71 43 73 41 84 30	1.00 4.55 ** 1.00 3.54 ** 1.00 5.48 ** 1.00 7.18 * 1.00 2.66 *	— 2.41–9.70 — 1.87–8.89 — 2.82–13.83 — 1.72–18.37 — 1.31–8.51
	Lower back (43.9%)	Deep bending Handling materials (>20 kg) Twisting the waist	≤20 times per day >20 times per day ≤20 times/per day >20 times/per day ≤20 times/per day >20 times/per day	68 46 71 43 65 49	1.00 4.23 * 1.00 3.186 * 1.00 4.51 **	— 1.31–13.66 — 1.83–11.47 — 1.95–10.43
	Neck (43.9%)	Sitting time Handling materials (>20 kg) Tobacco smoking	≤4 h per day >4 h per day ≤20 times/per day >20 times/per day No Yes	72 42 71 43 25 89	1.00 2.75 * 1.00 2.38 * 1.00 2.58 *	— 1.39–6.45 — 1.20–5.33 — 1.39–6.04
	Knees (36.8%)	Age Handling materials (>20 kg) Standing time	≤45 years >45 years ≤20 times/per day >20 times/per day ≤4 h per day >4 h per day	74 40 71 43 73 41	1.00 3.97 ** 1.00 2.03 * 1.00 2.74 *	— 2.01–8.74 — 1.67–6.55 — 1.21–6.83
Thai workers (*n* = 57)	Hands/wrists (42.1%)	Handling materials (>20 kg)	≤20 times/per day >20 times/per day	21 36	1.00 1.96 *	— 1.18–4.15
	Shoulders (36.8%)	Handling materials (>20 kg) Tobacco smoking	≤20 times/per day >20 times/per day No Yes	21 36 29 28	1.00 2.11 * 1.00 2.77 *	— 1.27–5.93 — 1.85–9.50
	Buttocks/legs (31.6%)	Standing time Carrying distance	≤4 h per day >4 h per day ≤5 m >5 m	6 51 29 28	1.00 7.82 * 1.00 3.89 **	— 1.20–40.52 0.09–0.39 1.30–13.54

Notes: * *p* < 0.05, ** *p* < 0.01; OR, odds ratio; CI, confidence interval.

## Data Availability

The data are available upon reasonable request to the Corresponding Author.

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
