# Peer review of "Comparative Ergonomic Study Examining the Work-Related Musculoskeletal Disorder Symptoms of Taiwanese and Thai Workers in a Tape Manufacturing Factory"

_ijerph, 2023, doi:10.3390/ijerph20042958_

Round 1
Reviewer 1 Report
Dear authors,
I really enjoyed reading your manuscript. You presented very interesting and valuable data that are difficult to gather. However I have some issues I hope you can clarify and I hope that my comments will help you improving the paper.
Methods: add details about the model of regression used
Line 235: when talking about discomfort in the preceding year, it is not clear to what you are referring
I think that the comparison with study [9] should be performed as Discussion part and not in the Results, because it makes the paragraph a bit confusing.
line 267: "administrative controls must 267 therefore be instituted" what does it mean? please specify
when comparing Thai and tawanese population of the study, why don't you perform a statistical test to see if the difference of prevalence is statistically significative? Also, can you say that the two population are very similar in terms of demographic, risk factors and baseline characteristics? Please specify.
Table 3, column "difference": it is not clear what does it mean and overall I don't believe it is useful, the difference between percentages is evident
Table 1 in my opinion suits better in Results.
Line 289- 291: "This implies that the two worker groups have different levels of physical fitness", honestly it does not imply at all. At the very most it may be different levels of physical fitness and this can have a role in the different prevalences of disturbs. Or specify better what you mean with "physical fitness"....maybe health in general?
It may be useful to add which is the limit weight for handling objects in Taiwan (25 kg? more?) prescribed by law and if there is any difference between age groups
Another limitation of the study is that all the disturbs are self reported and you don't have real diagnosis. THis can lead to overreporting for example.
Reviewer 2 Report
The paper can be accepted in its current form. However, the following explanations can improve the study:
How long have the workers been working at the factory? (NMQ can not be used for working periods less than 1 year).
The comments for the ILOSH data Table 3 can be related to the outcomes of the study.
The reasons for not using 3DSSPP for C and D tasks can be given.
Reviewer 3 Report
The authors conducted a study of the impact of human physical performance on the risk of WRMSD's. They provide a valuable source of information for the preparation of prevention plans. The method of presentation of results does not include much important information for the preparation of preventive actions. The method of presenting results according to RULA does not apply individually to key activities and segments of the musculoskeletal system, and does not include information on which the postures should be improved.
Detailed comments:
Page 4, lines 166-168: Using the RULA method, there should be an effort to reduce subjectivity by measuring the position angles of segments of the musculoskeletal system. Having multiple photographs of particular postures, an assessment can be made for each position, and by measuring the time taken to hold each position, the level of exposure to a given level of risk during work can be estimated.
Page 8, lines 263-265: I think it is worth specifying that these are not the true values of the compression forces, but the values assigned to the compression forces by applying the 3DSSPP model. The differences between the true value and the value read from the software, are due to components of measurement uncertainty. These include, for example, the quality of the digitization/modeling of postures and loads in the digital human model.
Page 8, Figure 2 (A and B): Based on measurements of the position angles of the musculoskeletal segments, it can be concluded that the postures mapped in the digital human model are far from the postures presented in the photographs. This affects the values and exposure ratings.
Page 9, lines 275-277: How the postures of the various segments of the musculoskeletal system were measured. The description of how the RULA and OWAS methods were applied is not very detailed. I think it should be elaborated on, supplementing it with assessment steps.
Page 10, lines 369-371: These conclusions were given, based on analysis of models in the early 1990s. Can more recent analyses be cited? It was not mentioned that also the quality of the mapping of positions in the models can also influence on the results.
